# Investigation of Newly Developed PCM/SiC Composite Aggregate to Improve Residual Performance after Exposure to High Temperature

**DOI:** 10.3390/ma15051959

**Published:** 2022-03-07

**Authors:** Dong Ho Yoo, Jeong Bae Lee, Hyunseok Lee, Hong Gi Kim

**Affiliations:** 1Civil and Environmental Engineering Department, Hanyang University, Jaesung Civil Engineering Building, 222 Wangsimni-ro, Seongdong Gu, Seoul 04763, Korea; dongho3461@naver.com; 2Department of Civil Engineering, Daejin University, 1007, Hoguk-ro, Pocheon-si 11159, Korea; dlwjdqo@nate.com; 3Division of Horticulture and Medicinal Plant, College of Life Science and Biotechnology, Andong National University, Andong 36729, Korea

**Keywords:** concrete, phase change material, silicon carbide, residual performance, fire-resistance

## Abstract

High temperature conditions, such as fire, are detrimental factors to the mechanical and chemical properties of concrete. In this paper, the authors developed a new type of coarse aggregate, named PCM/SiC composite aggregate (enhanced aggregate: EA), to improve fire-resistance performance. To investigate the validity of EA for construction materials, a compressive strength test, static modulus of elasticity, X-ray diffraction (XRD), and scanning electron microscopy (SEM) were conducted. In addition, this EA has been developed to improve residual performance after exposure to high temperature, with residual compressive strength and internal temperature measurement tested at 1000 °C. Furthermore, chemical properties after heating were investigated by XRD and SEM-EDAX. The results show that the percentage of residual compressive strength of heated concrete with EA is higher than plain concrete. The concrete with EA exhibited primary cement composites such as C-H and C-S-H after exposure to high temperature through XRD and SEM-EDAX. On the other hand, major hydration products could not be observed in plain concrete. PCM and SiC offer an opportunity to delay the increase in concrete temperature. From evaluation of the results, we can see that EA enhanced the residual performance of concrete after exposure to high temperature conditions.

## 1. Introduction

Concrete structures are incombustible construction materials compared with wood and steel when exposed to fire or high-level temperatures. Though concrete is a nonflammable material, the mechanical, chemical and durability properties of concrete can be detrimental at elevated temperature conditions [1]. These properties of concrete after exposure to high temperature are generally known as their residual properties. Residual properties are determined by the binder, aggregate and water that compose the concrete. Furthermore, type of fire load [2], aggregate type and size, type of cement/binder paste, and water/cement ratio [3] can all be influenced by the way the concrete’s properties change. In order to improve on specific properties after exposure to elevated temperatures, one can vary the material composition of the concrete [4], such as its supplementary cementitious materials (SCMs), fibers, and aggregates types [5].

There is previous research that has studied the thermal performance of concrete blended with SCMs, such as fly ash (FA), silica fume (SF), and grounded granulated blast furnace slag (GGBS) to improve the properties of concrete after exposure to high temperatures. Pozzolanic concrete with fly ash (30%) and blast furnace slag (40%) have shown a higher retained residual strength in normal concrete at high temperatures of around 600 °C [6]. Also, Bastami et al. [7] have evaluated high-strength concrete containing nano silica, they observed that the critical temperature level for domestic compressive strength loss was increased from 400–800 °C to 600–800 °C by adding the nano silica.

In addition, many researchers have investigated the thermal performance of concrete when adding diverse fiber types at elevated temperatures. Lau et al. [8] compared the compressive strength of steel fiber-reinforced concrete (SFRC) with 1% addition vs. non-SFRC. The results show that the compressive strength loss of SFRC at high temperature (below 1000 °C) was significantly slower than non-SFRC. In addition, polypropylene (PP) fiber also increases the residual properties of concrete. Eidan et al. [9] have confirmed that PP fiber-reinforced concrete has improved residual mechanical strength compared with plain concrete, as the PP fiber neutralizes the effects of the physiochemical degradation of cement above 400 °C.

Finally, the impact of aggregate type of residual performance on concrete has been studied by various researchers. Generally, natural aggregate has poor performance in residual compressive strength compared with recycled concrete aggregate (RCA) at high temperature because natural aggregate’s composition of siliceous mineral expands at elevated temperature. Salahuddin et al. [10] investigated the residual compressive strength of 100% RCA and control mix of natural aggregate. The results show that at a high temperature of 600 °C, the compressive strength of 100% RCA concrete slightly improved compared with control concrete due to the increased amount of recycled mortar attached to the recycled aggregate, which enhances the thermal expansion properties inside of the concrete.

Research on improving residual mechanical properties in the above methods is being actively conducted. However, little is known about the new type of developed aggregate. For this reason, this study developed a new type of aggregate to improve residual performance after exposure to high temperatures. The new type of developed composite aggregate in this research was called enforced aggregate (EA), which refers to PCM/SiC composite aggregates. This EA was used for coarse aggregate in concrete, while concrete that contains EA shows a delay in the time it takes to increase the internal temperature when exposed to elevated temperatures. Therefore, first of all, the mechanical and chemical properties of concrete with EA at normal conditions were examined to evaluate EA’s usefulness as a construction material. This was undertaken through compressive strength test, X-ray diffraction (XRD), scanning electron microscopy and energy dispersive spectroscopy (SEM-EDAX). In addition, to investigate the effect of EA on the behavior of concrete at high temperatures, residual compressive strength was examined after various high temperature and internal temperature measurements of concrete with EA were conducted in an electric furnace at 1000 °C. XRD and SEM-EDAX were performed to explore chemical properties after exposure high temperature.

## 2. Materials and Methods

### 2.1. Materials and Fabrication of PCM/SiC Composites Aggregate

This study used ordinary Portland cement (OPC), which follows ASTM C150 [11] as the cementitious material. Natural sand (maximum size: 4.75 mm, fineness modulus: 2.57, density 2.62 ton/m^3^, water absorption: 1.11%) was used as fine aggregate. Paraffin wax and silicon carbide (SiC) were used for developing the high thermal resistance aggregate as impregnating material and coating, respectively. SiC (SiC > 94%, Fe_2_O_3_ < 0.7%, Fe_3_C < 0.5%) was obtained from a local company (Hansong Co., Hwaseong, Korea). High porosity of air-cooled blast furnace slag aggregate (ACBFS) are selected as a phase-change material (PCM) carrier which was offered by manufacturer (Hyoseok Co., Gwangyang, Korea) (Gmax: 25 mm, absolute dry specific gravity: 2.59 g/cm^3^_,_ fineness modulus: 3.12). Paraffin wax produced by the manufacturer (Nippon-Seiro Co., Kyobashi, Japan) is well known for having a mixture composed of hydrogen and carbon molecules with the configuration (C_n_H_2n+2_). It has a high specific fusion heat, is reliable, less expansive and is chemically stable when it is flashed. Paraffin wax has a melting point of 47 °C and a flash point of 202 °C.

The processes of coating and impregnation were conducted as follows;

(1)Prepared ACBFS in the bowl with melted paraffin wax at 80 °C in dry oven for 4 h at 100 °C.(2)After 4 h, the dredged impregnated slag aggregates are put in an ice chamber quickly and left to congeal for 10 min.(3)After the prior steps, SiC is coated on the surface of the impregnated aggregates with mixed polymer and PVA water soluble film.

Figure 1 shows a more detailed overview of the manufacture of EA.

### 2.2. Mix Proportions

The concrete mixes were designed with a target strength of 30 Mpa [12]. The slump and air content were 180 mm and 4.5% respectively. Additionally, water-binder ratio for 44.9%, sand-coarse aggregate ratio for 45.8%, and water reducer-binder ratio for 0.5% were selected. In this study, two kinds of the concrete mixture were prepared. In one case, the EAs were replaced 30, 50, 70 and 100% with the natural coarse aggregates (maximum size: 25 mm, fineness modulus: 6.91, density 2.66 ton/m^3^). In the other case, 5% of the SiC was replaced by the natural fine aggregate which, with its finer particle size compared with normal fine aggregate, could improve the mechanical properties. Then, these concretes were evaluated for their mechanical and chemical properties and their post-fire residual properties. Table 1 shows the properties of the concrete mixture.

### 2.3. Experimental Methods

#### 2.3.1. Mechanical and Chemical Properties of the Concrete with PCM/SiC-Based Composite Aggregate

Specimens were prepared with 100 mm (diameter) × 200 mm (height) cylinder concretes to perform compressive strength tests after 3, 7, and 28 days as per ASTM C39 [14]. In addition, static modulus of elasticity test was conducted by ASTM C 469-94 [15] for evaluating the mechanical properties of concrete with EA. To analyze the hydration products of samples with EAs, SEM-EDAX, and XRD were conducted. In addition, EDAX analysis was conducted if a verification of chemical components by samples was needed. Samples for SEM were manufactured according to the method in Petrographic Methods of Examining Hardened Concrete: A Petrographic Manual. To analyze the microstructure of a substance according to diffracting X-rays, XRD was conducted. The power was packed into sample holders for use in a Rigaku X-ray Diffractometer by CuK radiation, operating at a voltage of 30 Kv, and a current of 20 Ma. A scanning speed of 2°/min and a step size of 0.01° were used to examine the samples in the range of 5~25° (2θ) to cover the phases under investigation.

#### 2.3.2. Residual Performance Properties at High Temperature for Concrete with PCM/SiC-Based Composite Aggregate

In the case of concrete, physical and mechanical properties are very important when it is exposed to high temperature. In particular, concrete which is exposed to high temperature undergoes severe damage and a change of its chemical composition. These changes usually result in the generation of a hardened cement paste by the dissociation of calcium hydroxide (C-H) at 400 °C. Afterward, it continues toward a complete decomposition and destruction of the hydration products, such as calcium silicate hydrate (C-S-H) gel, at around 900 °C. Furthermore, the effects of exposure to the high temperature could be revealed in the forms of cracking, spalling and disintegration on the surface which is destructive to the concrete structure. Therefore, residual strength is a considerably important factor for evaluating the deterioration degree of the concrete. The residual compressive strength test was conducted after exposure to temperatures of 200, 400, 600 and 800 °C. Specimens were taken out of the furnace and then a compressive strength test was conducted after cooling, upon reaching each temperature interval. In the cases of those exposed to high temperature, the internal temperature condition of the concrete is significantly important point in terms of its structural safety. The use of PCM in concrete can improve its high thermal resistance. The PCM hinders the increasing temperature of concrete because it absorbs heat during two steps of their phase changing, solid to liquid (around melting point) and liquid to gas (around flash point) [16]. Therefore, an internal temperature measurement was conducted with the furnace temperature set at 1000 °C. The cylindrical mold was used for internal temperature measurement and K-type thermocouples were inserted into the center of the specimens. Figure 2 shows an example of the appearance of specimens in the electric furnace before and after experiment. The color change was almost similar in each of the specimens. Additionally, XRD and SEM-EDAX analysis was conducted with samples prepared after high temperature exposure in order to verify residual properties.

## 3. Results and Discussion

### 3.1. Mechanical and Chemical Properties of Concrete with PCM/SiC Composite Aggregates

#### 3.1.1. Compressive Strength Test

To evaluate the performance of the physical properties of specimens with EAs, a compressive strength test was conducted after 3, 7 and 28 days. The results of the compressive strength tests are shown in Figure 3.

From the results, the compressive strength values of the specimens EAs were decreased in comparison with the plain specimen. Plain specimen was exposed to 26.78 MPa after 3 days. Finally, it was exposed to 41.04 and 46.24 after 7 and 28 days, respectively. However, the specimen with 50% of EA was exposed to 21.2, 30.4 and 36.7 MPa after 3, 7 and 28 days respectively, reflecting a decrease of 21, 27 and 21% in comparison with the plain specimen after 3, 7 and 28 days respectively. In addition, the specimen composed of 100% EA showed a decrease of 19, 18 and 19% in comparison with plain specimen after 3, 7 and 28 days respectively. The compressive strength decreased slightly in comparison with the plain specimen with increased usage of EAs. On the other hand, the P+SiC specimen demonstrated a compressive strength of 28.54 after 3 days, while after 7 and 28 days, was at 43.12 and 49.27 respectively. The compressive strength of the EA100+SiC specimen reached 23.5, 36.4 and 40.71 MPa after 3,7 and 28 days, respectively, which was lower than P+SiC by about 21%. Thus, compressive strength decreased with the increased usage of EAs. The reduction of this mechanical property is caused by comprehensive effects. In this study, the coarse aggregate was replaced with EAs [17]. Therefore, the adhesive properties between the paste and EA could cause a weaker than natural aggregate. In addition, air gaps between the paste and the aggregate, known as the interfacial transition zone (ITZ), may affect the mechanical properties, causing their reduction [18,19]. SiC has much smaller size of particle than fine aggregate. Thus, it could lead to more significant bridging effects and the formation of voids after hydration [20]. Therefore, the compressive strengths of specimens with SiC were improved in comparison with specimens without SiC. However, low density and high porosity in the concrete had negative influence on its mechanical properties, such as compressive strength and durability [21]. Therefore, we can attribute to specimens with EA a reduction of compressive strength compared with specimens with natural aggregate. However, specimens with EA offered satisfactory results for the design-strength value. Thus, EA can be used as materials for concrete structures.

#### 3.1.2. Static Modulus of Elasticity

From the results of the stress-and-strain curves in Figure 4 and Figure 5, the curvilinear stress-and-strain curves for the specimens indicate the largest difference associated with the behavior of specimens under short-term loading. The total of 6 stress-and-strain curves indicate a repeatable characteristic driven by the influence of the aggregate types and ITZ properties within the replacement ratio.

The highest vulnerability of the deformation was observed under a short-term loading with each replacement ratio of EAs. Furthermore, the highest effect of the deformation under load is greatly related to specimens with a low modulus of elasticity. It is assumed that the lower adhesion of EA compared with natural aggregate with cement paste can be attributed to the oil contents of paraffin wax.

As mentioned in the compressive strength section, the poor adhesion between paste and EAs could be attributed to the reduction of the mechanical properties. On the other hand, the use of SiC is slightly affected by the improvement of the mechanical properties of the deformation. The static modulus of elasticity of the specimens with SiC was increased in comparison with those without SiC. However, the reduction of static modulus of elasticity in the EA100 and EA100+SiC specimens was slightly minor. Therefore, the use of EA is reasonable to be applied in concrete structures. 

#### 3.1.3. XRD Analysis

To evaluate hydration characteristic and chemical composition of specimens with EAs, XRD patterns were analyzed after 3, 7 and 28 days. The XRD patterns of hydrated samples are shown in Figure 6, Figure 7 and Figure 8.

The clinker materials can be seen in Figure 8. These peaks tended to decrease over the time of hydration up to 28 days. Ettringite peak in concrete samples gradually decreases according to their curing times. In addition, samples with SiC indicated that their peak is located at approximately 35.8°, which is shown in each Figure with SiC addition. SiC is not a hydration reactive material so the main hydration components, such as C-S-H and CH, are similar for each XRD, except SiC peak [22].

#### 3.1.4. SEM-EDAX Analysis before Exposure to Heat

To analyze the hydration products of specimens with or without EAs, SEM-EDAX analysis was conducted after 3, 7 and 28 days. The results are shown in Figure 9, Figure 10, Figure 11, Figure 12, Figure 13 and Figure 14.

Various hydrated phases were observed in all figures, such as C-H, Ettringite, and C-S-H formation. These products are commonly observed in plain and EA samples after 3 days. In case of samples with SiC, C-H and ettringite formation was also observed after 3 days. The phases are much denser after 7 days, with C-H formation in particular much bigger than at 3 days.

However, C-S-H was observed around SiC particles in P+SiC and EA100+SiC samples after 7 days. SiC particles were verified by EDAX analysis. As per the results of the EDAX analysis, it can be confirmed that the peak of O (oxygen) is very low. In addition, it can be confirmed that paste and SiC are well combined in the matrix.

In addition, by day 28, the structure was considerably denser than at 7 days. Furthermore, the Si peak of the EA100 sample was lower than in the plain sample per the results obtained by the EDAX analysis. The effect of paraffin wax on hydration can be observed by the mass of paraffin wax lost to hydration. Therefore, this can be affected to reduce the hydration products at the initial stage. This result can be verified with the results of EDAX for the EA100+SiC sample in comparison with the P+SiC sample. All chemical elements are lower than P+SiC. However, SiC particles are well combined in the cement paste. Thus, it can be proved that the mechanical properties were improved with the use of SiC as a filler.

### 3.2. Thermal Resistant Properties at High Temperature of Concrete with PSA

#### 3.2.1. Observation on the Surface of the Specimens after High-Temperature Exposure

To confirm the condition on the surface of samples after high-temperature exposure (1000 °C), the damaging effect on the surface of samples was observed by microscope, 160 times magnification. The results of the observation on the samples are shown in Figure 15.

From the results, a lot of cracks on the surface of the plain sample can be observed after high-temperature exposure. Verification of the concrete damaged by high temperature can be seen with visual observation of the color changes [23], as well as cracks and spalling. The plain specimen was completely decomposed and lost binding properties at high-temperature exposure to 1000 °C. Furthermore, excessive spalling on the surface of the sample was observed, which was caused by constant crack formation after 800 °C [24].

In the case of the heated concrete surface, the damage was significantly affected by parallel degradation of the concrete strength and pressure of concrete pores [25]. In addition, the explosion of the thermal spalling is attributed to the hidden breaking of the crushed concrete [26]. As can be seen in Figure 15, the depth of cracks on the surface of the plain sample were much higher than for the EA100 and EA100+SiC samples. Furthermore, the spalling condition can be verified in Figure 15. However, the cracks of the EA samples were lower than the plain sample and the color was brighter.

SiC particles (green) around aggregate can be observed in Figure 15. As with a previous study, it is demonstrated that the samples with SiC have a significant fire resistance after high-temperature exposure at 1200 °C [27], with the weight loss associated with the SiC at 1000 °C not being generated [28]. SiC particles were coated on the surface of paraffin coating, when manufacturing EAs, which shows that SiC on the surface of the aggregate or around ITZ may be protected from the decomposition or dehydration processes caused by high temperature exposure.

#### 3.2.2. Residual Strength Test

Mechanical and physical properties are dramatically decreased with high temperature exposure. Therefore, residual compressive strength tests were conducted after various temperature (200, 400, 600 and 800 °C) exposures. Specimens were cooled at room temperature for this test. Figure 16 shows the results of residual rates of compressive strength.

These were calculated by using the percentage of retained strength in comparison with unheated specimens. The results show that the residual rate of compressive strength on every specimen was significantly decreased. Residual rate of the compressive strength of the plain specimen was 33.4% at 800 °C. In the case of the P+SiC specimen, this was at 35.2% after exposure to 800 °C. However, the EA100 and EA100+SiC specimens were higher than the plain and P+SiC specimens. The EA100+SiC specimen was at 56.24% after being exposed to 800 °C. The compressive strength could be reduced by the dehydration caused by evaporating free water due to the high temperature [25,27,29]. Furthermore, compressive strength reduction can be caused by expansion of the aggregate at high temperature [24]. Therefore, the damage by high temperature on the residual compressive strength is considerably related to the type of aggregates in the concrete system. However, paraffin wax has a flashpoint of 200 °C, which can hinder aggregate expansion due to the paraffin wax absorbing the heat instead of the aggregate. As mentioned above in Section 3.2.1, SiC is very stable at high temperatures. It maintains a bridging effect after being exposed at high temperature. Therefore, specimens with EA showed a reduction of compressive strength that was lower than the plain specimen.

#### 3.2.3. Internal Temperature Test at High-Temperature Exposure

The results of internal temperature measurements of the specimens are shown in Figure 17, Figure 18 and Figure 19. As shown in Figure 17 and Figure 18, the temperature movement of the plain specimen gradually increased smoothly without a specific change. However, the specimens containing EAs display a resistance to the temperature increase in a certain section. To be specific, they show a very slightly increase from 21 °C to 140 °C. This is due to the fact that paraffin wax has a phase-change temperature of around 40 °C and flashpoint of 200 °C, so paraffin wax intercepts heat for its phase change and suppresses the rise in temperature inside the concrete.

In Figure 19, the EA100+SiC specimen in particular shows a much lower temperature behavior than that of the EA100 specimen. This phenomenon was retained for 16 min. The increasing temperature trends were delayed around 200 °C for 10 to 13 min in concrete with EA. Afterward, the temperature smoothly increased to 1000 °C as with the plain specimen. Hydration products lose free water after which they start to lose bonded water chemically from 105 °C [4,30]. In case of capillary water, this is completely lost at 400 °C [31]. In addition, siliceous aggregates are transformed due to the expansion of the concrete [23]. Paraffin wax decomposes at around 200 °C. Regarding the processes in the specimen with EAs, such as decomposition, dehydration and the melting of paraffin wax, these can delay the increase of the temperature at the initial stage due to the characteristics of SiC and paraffin wax when exposed to high temperature. Thus, paraffin wax can absorb heat during an elevated temperature condition. This phenomenon can be attributed to a delay in the increase of temperature for approximately 9 min. As for the result, the delayed time of temperature increase in comparison with normal aggregate can be attributed to EA. Therefore, the addition of EAs can improve the fire-resistance performance of a concrete structure.

#### 3.2.4. XRD Analysis

In order to evaluate the chemical analysis of the samples after heating, an XRD analysis was conducted. The results of the XRD analysis of the heated samples are shown in Figure 20.

From the results, C-S-H intensity was not observed in the plain sample. In addition, intensity of C_3_S and C_2_S is higher than EA samples in the results. The results show that C-S-H phase decomposed due to the high temperature at around 800 °C. At that time, the intensity of C_3_S and C_2_S increased whereas the intensity of C-S-H decreased. The C-S-H intensity of the plain sample was not observed, whereas C-S-H intensity of EA100 sample was observed around 28°. On the other hand, the intensity of the paraffin wax was indicated in the EA samples. Carbon intensity was observed at around 27° because of the paraffin wax.

In addition, SiC intensity was observed in the EA samples at around 36°. Every sample for XRD analysis was produced after internal temperature experiment was conducted to 1000 °C. Thus, SiC intensity can be seen in the result. In addition, paraffin wax remained in the samples with EAs after high-temperature exposure. As result, paraffin wax and SiC on the surface of the aggregate can affected a much lower decomposition and dehydration around the ITZ parts. Thus, EAs can prevent rapid reduction of the mechanical and physical properties in a concrete system at high-temperatures exposure.

#### 3.2.5. SEM-EDAX Analysis after Heat Exposure 

The microstructure and physical state of heated samples was analyzed by SEM-EDAX. The results of SEM-EDAX are shown in Figure 21, Figure 22, Figure 23 and Figure 24.

The results show that the C-S-H phases disappeared. Also, many cracks can be observed in Figure 21. In addition, decomposed parts can be seen in Figure 22. This is caused by decomposition or dehydration of hydrated products, weakening them.

On the other hand, the EA100 sample showed the C-S-H and CH products that can be observed in Figure 23. Also, the structure of the hydration products was much denser than the plain and P+SiC samples. In addition, decomposition and dehydration parts can be observed in Figure 24.

However, EA100 sample can be seen with C-S-H gels and CH. As shown by the results of the plain and P+SiC EDAX analyses, the peak and weight of Ca is much higher than in the EA100 sample. However, in the case of the Si peak and weight, EA100 and EA100+SiC samples are much higher than the P and P+SiC samples. In the case of the EA100 and EA100+SiC samples, SiC particles can be verified around hydration products. As shown by the results of the EDAX analysis, there are Si and C peaks. Furthermore, the O peak is much lower than the others. Therefore, it can be proved that SiC particles remained after high-temperature exposure. Thus, SiC particles on the surface of the aggregate can be attributed to a delay of, and protection against, the decomposition and dehydration processes. Finally, the concrete with EA has demonstrated that it has a good ability to improve fire resistance. These results demonstrate that a delay in the decomposition time and decrease of strength can be attributed to paraffin wax and SiC coating.

## 4. Conclusions

The main purpose of this study was to develop a PCM/SiC composite aggregate called EA. The properties of concretes containing EA were investigated before and after exposure to high temperatures.

Properties of concrete before exposure to high temperatures.
The compressive strength of concrete with EA reduces gradually according to the increasing replacement of natural aggregate with EA. However, compressive strength tended to increase by adding SiC whose particle size is smaller than the fine aggregate. SiC can lead to a bridging effect and the formation of voids after hydration.The reason for the deformation under load is highly related to the low modulus of elasticity of specimens. This shows that poor adhesion of EA between the surface of EA and cement paste can be attributed to the oil contents of paraffin wax. However, SiC is slightly affected by the improvement of the mechanical properties of the deformation.From the XRD results, the tendency of plain concrete and concrete containing EA are similar because PCM and SiC are non-reactive materials.In the SEM-EDAX section, main gradients in concrete, such as C-H and C-S-H were observed. Also, SiC particles were well bonded to the cement matrix, which can improve its mechanical properties.Properties of concrete after exposure to high temperatures.
EA100+SiC has the lowest reduction in compressive strength after high-temperature exposure. This is because paraffin wax absorbs the heat that would otherwise power aggregate expansion and SiC maintains the bridge effect even after heating.According to the internal temperature measurement test, the temperature behavior of EA100 and EA100+SiC specimens show a lower temperature trend compared with plain concrete. Also, the EA100+SiC specimen was delayed by 16 min in reaching 200 °C, which means that EAs in concrete hamper the increasing temperature at the initial stage in elevated temperature conditions.In SEM-EDAX, after exposure to high temperature, C-S-H phases disappeared and there were easily observed cracks. However, the concrete specimen with EA displayed S-C-H gels and C-H. Delay and protection against the decomposition of chemical products can be attributed to EA.

## Figures and Tables

**Figure 1 materials-15-01959-f001:**
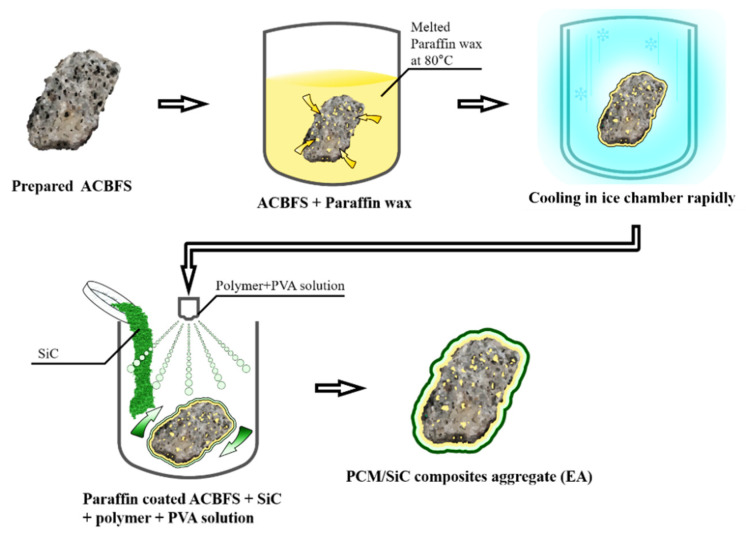
Schematic fabrication of PCM/SiC composites aggregate.

**Figure 2 materials-15-01959-f002:**
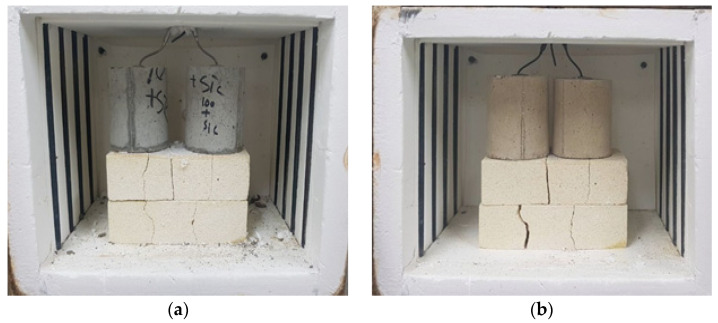
Specimens in the electric furnace. (**a**) before heating at 1000 °C (**b**) after heating 1000 °C.

**Figure 3 materials-15-01959-f003:**
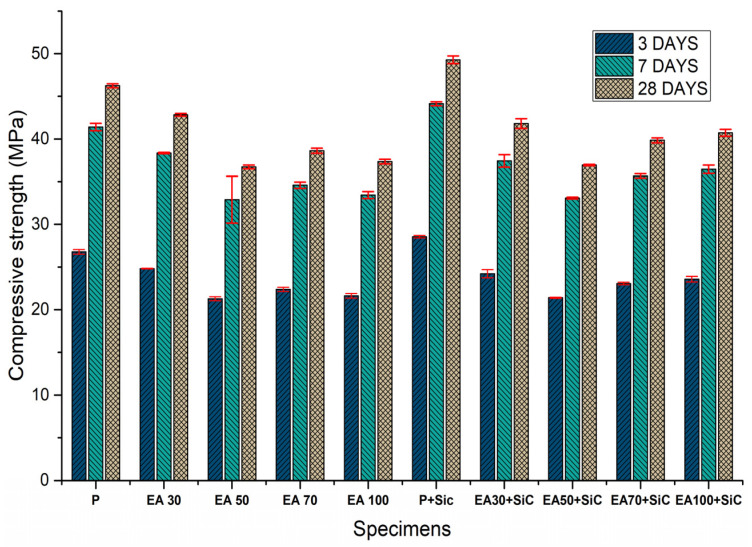
Results of the compressive strength tests after 28 days.

**Figure 4 materials-15-01959-f004:**
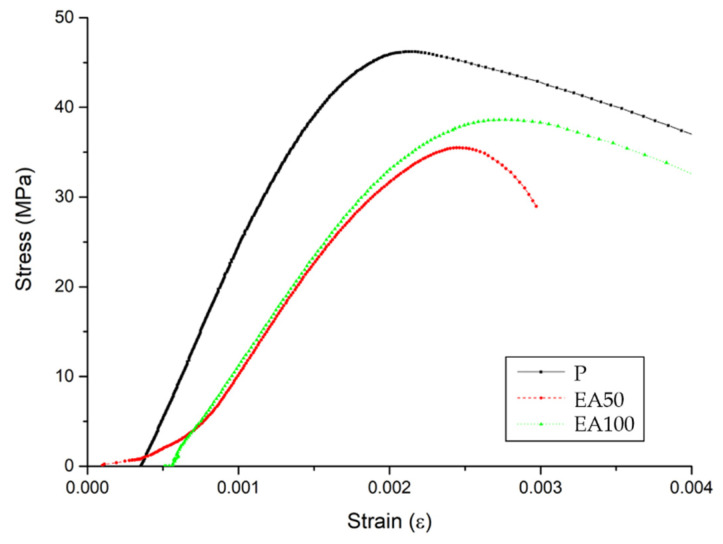
The results of the stress/strain curve after 28 days (without SiC).

**Figure 5 materials-15-01959-f005:**
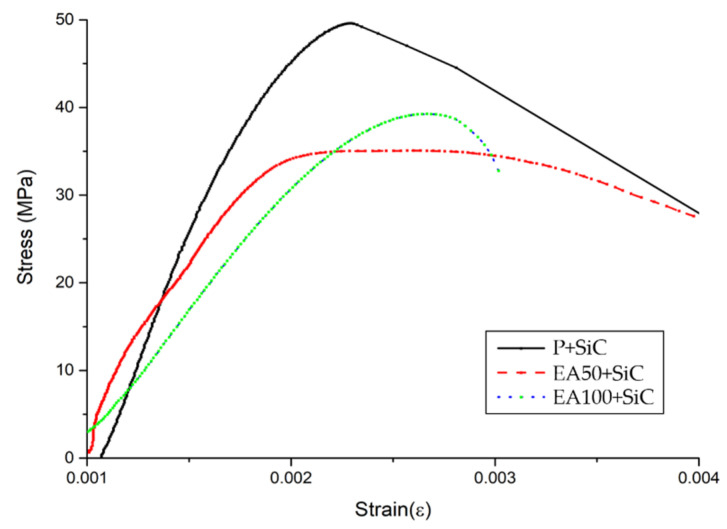
The results of the stress/strain curve after 28 days (with SiC).

**Figure 6 materials-15-01959-f006:**
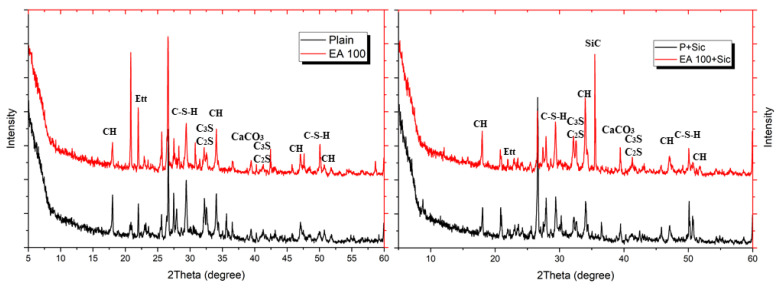
Results of XRD after 3 days.

**Figure 7 materials-15-01959-f007:**
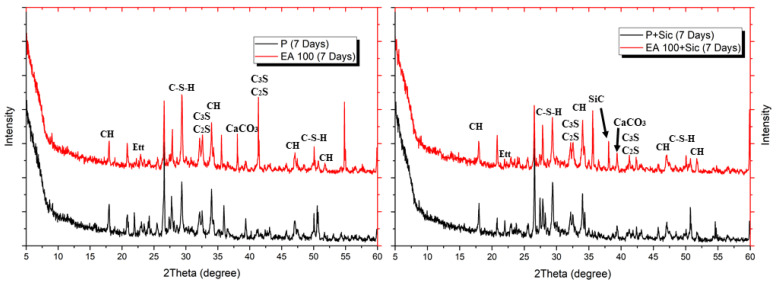
Results of XRD after 7 days.

**Figure 8 materials-15-01959-f008:**
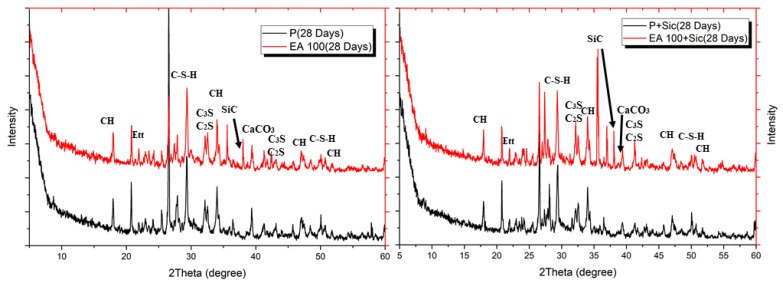
Results of XRD after 28 days.

**Figure 9 materials-15-01959-f009:**
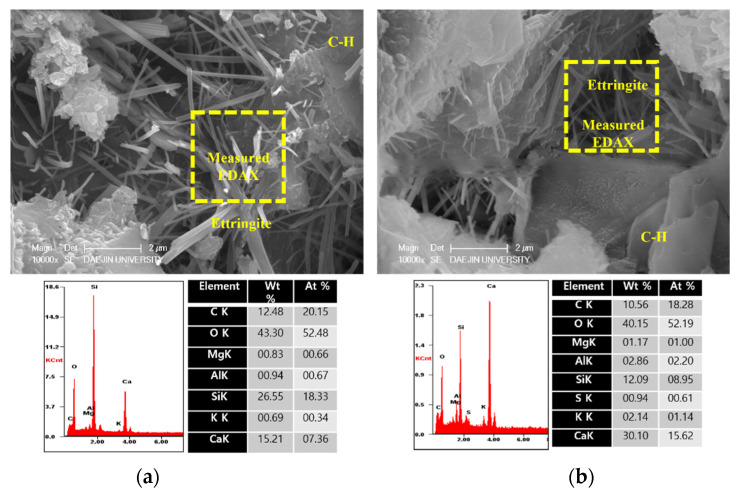
SEM Images with E-DAX analysis after 3 days ((**a**); plain, (**b**); EA100). Reprinted from [12].

**Figure 10 materials-15-01959-f010:**
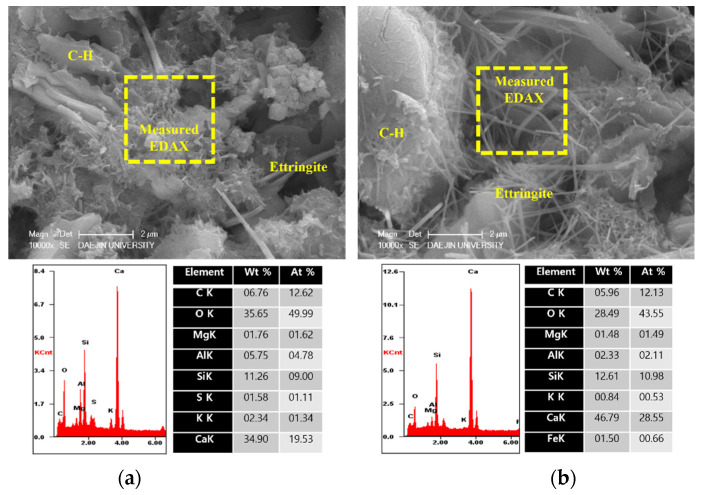
SEM Images with E-DAX analysis after 3 days ((**a**); plain + SiC, (**b**); EA100+SiC). Reprinted from [12].

**Figure 11 materials-15-01959-f011:**
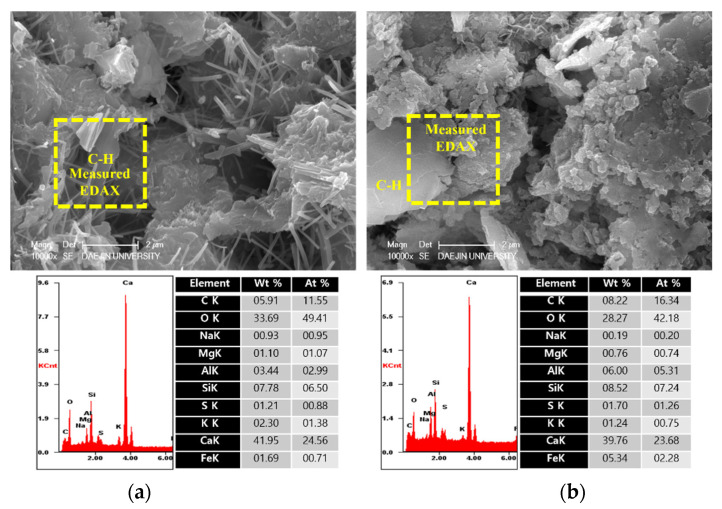
SEM Images with E-DAX analysis after 7 days ((**a**); plain, (**b**); EA100). Reprinted from [12].

**Figure 12 materials-15-01959-f012:**
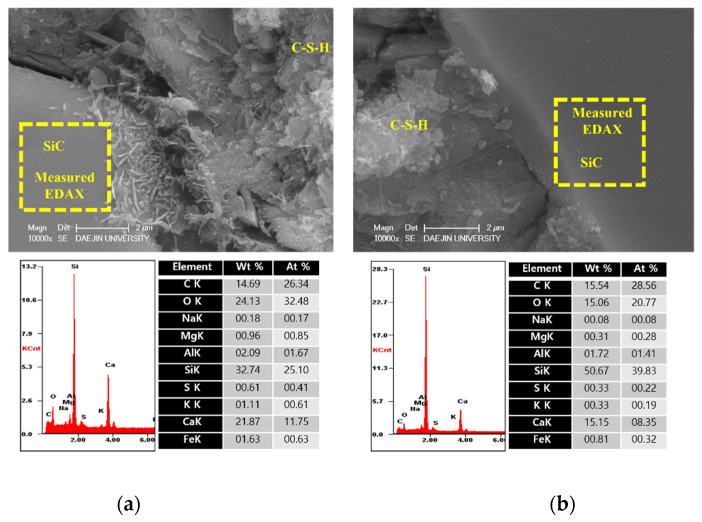
SEM Images with E-DAX analysis after 7 days ((**a**); plain+SiC, (**b**); EA100+SiC). Reprinted from [12].

**Figure 13 materials-15-01959-f013:**
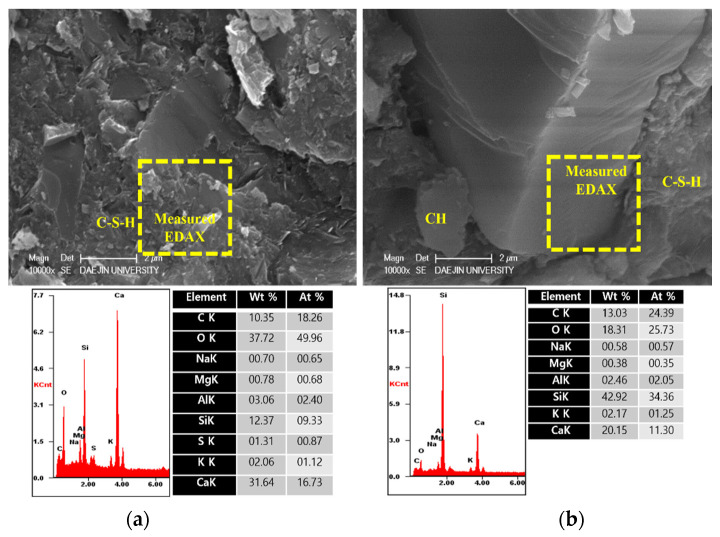
SEM Images with E-DAX analysis after 28 days ((**a**); Plain, (**b**); EA100). Reprinted from [12].

**Figure 14 materials-15-01959-f014:**
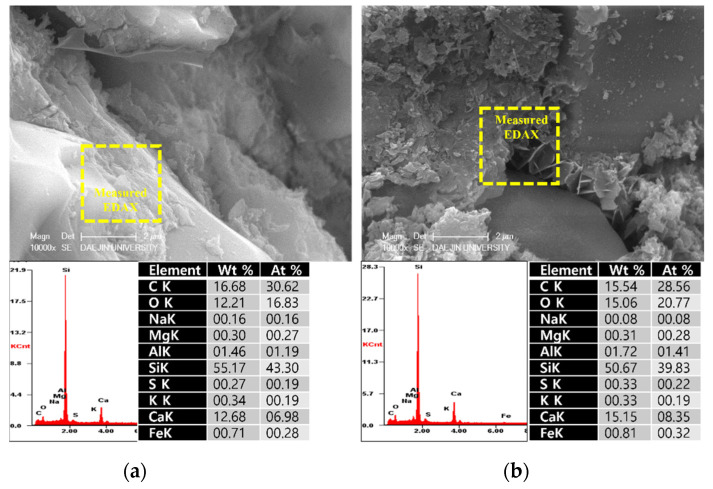
SEM Images with E-DAX analysis after 28 days ((**a**); Plain+SiC, (**b**); EA100+SiC). Reprinted from [12].

**Figure 15 materials-15-01959-f015:**
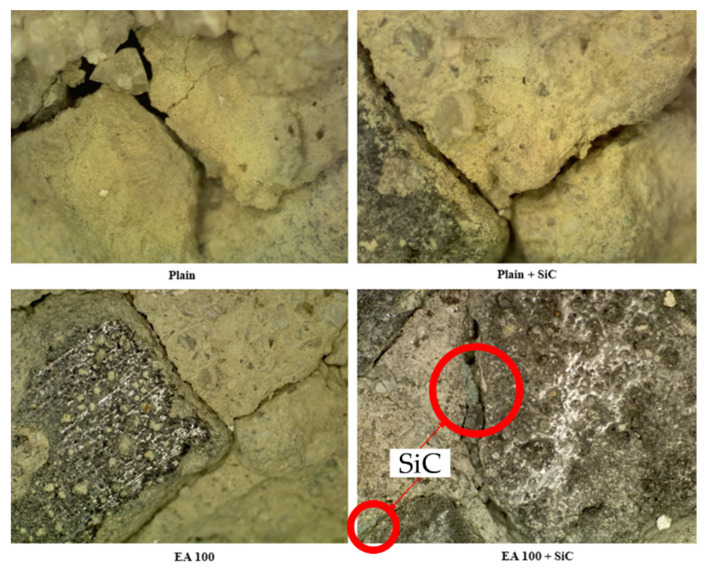
Observation on the surface of samples after internal temperature experiment conducted at 1000 °C.

**Figure 16 materials-15-01959-f016:**
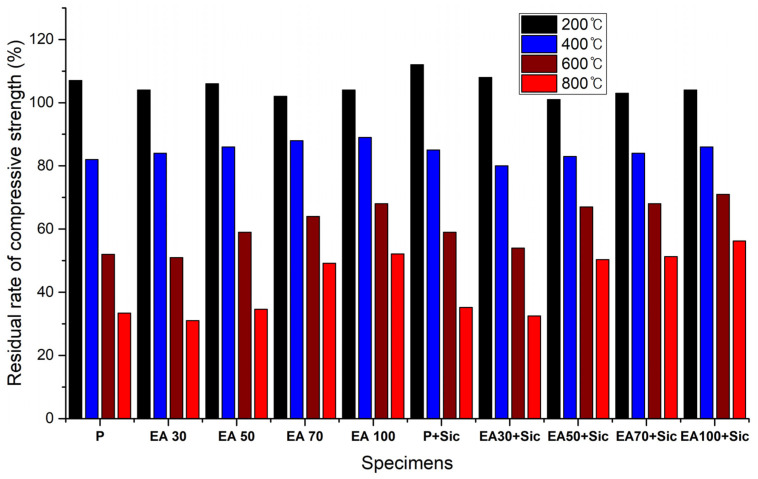
Results of the residual rates of compressive strength at 28 days (%).

**Figure 17 materials-15-01959-f017:**
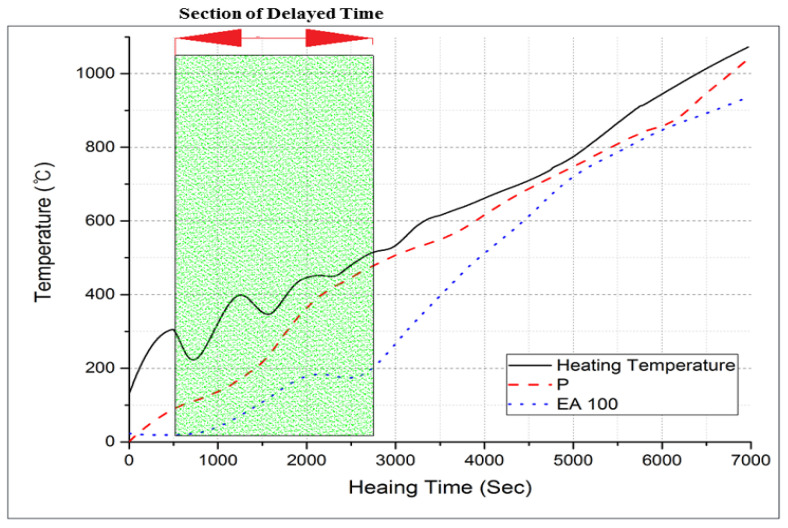
Internal temperature experiment results from 0 to 1000 °C (P and EA100 specimen).

**Figure 18 materials-15-01959-f018:**
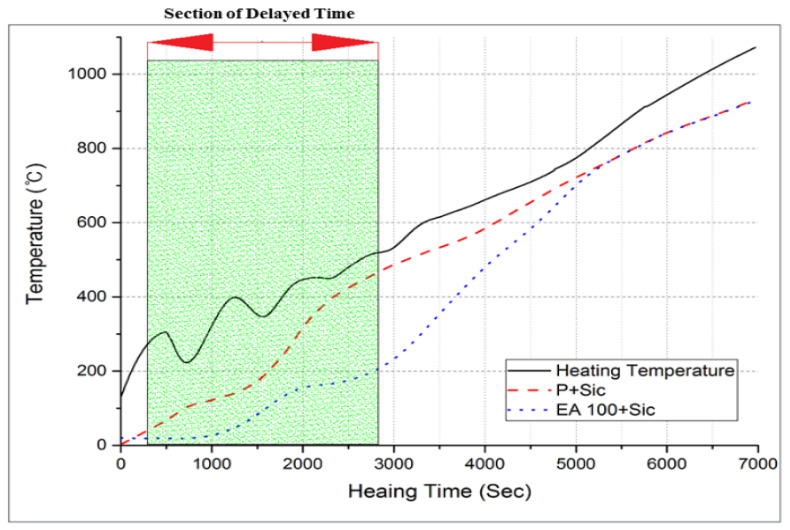
Internal temperature experiment results from 0 to 1000 °C (P+SiC and EA100+SiC specimen).

**Figure 19 materials-15-01959-f019:**
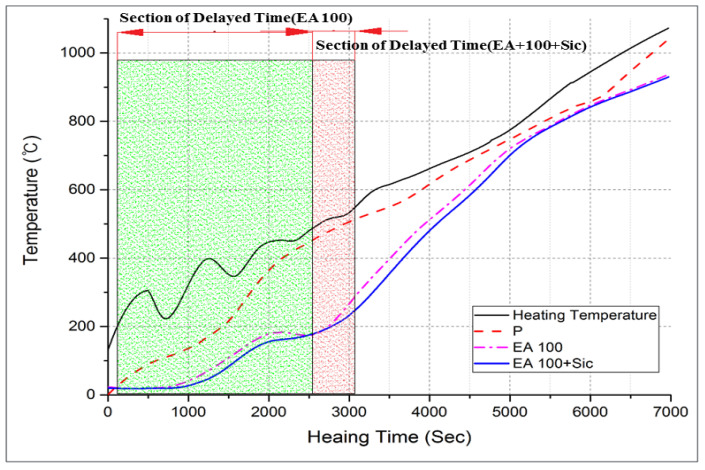
Internal temperature experiment results from 0 to 1000 °C (P, EA100 and EA100+SiC specimen).

**Figure 20 materials-15-01959-f020:**
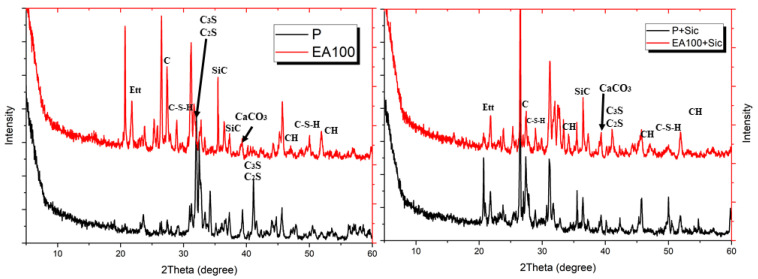
XRD patterns of heated samples after 28 days.

**Figure 21 materials-15-01959-f021:**
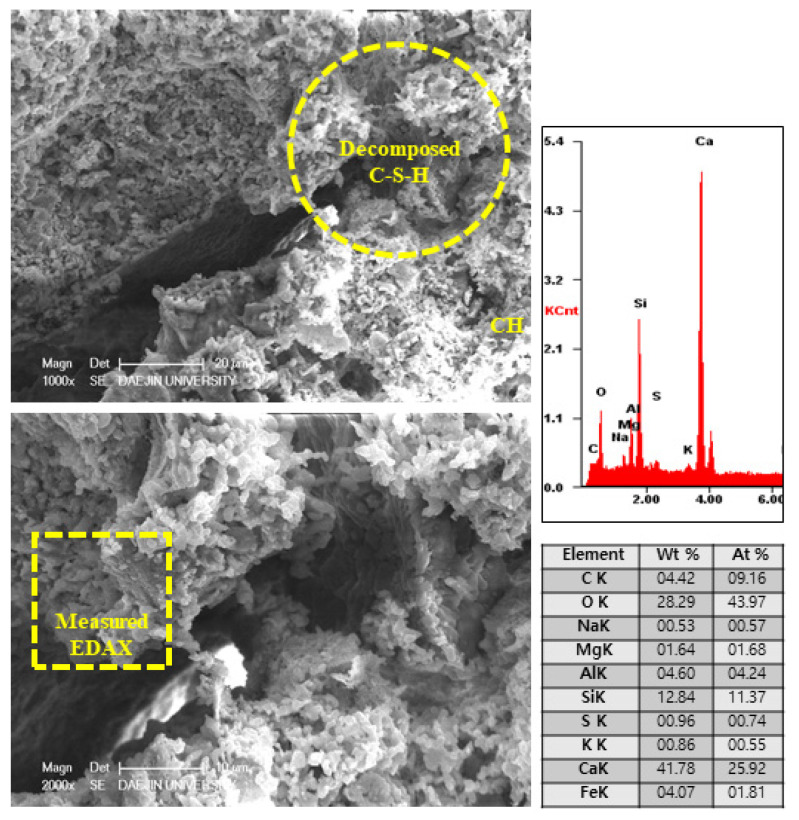
SEM-EDAX results of the plain sample after internal temperature experiment (1000 °C).

**Figure 22 materials-15-01959-f022:**
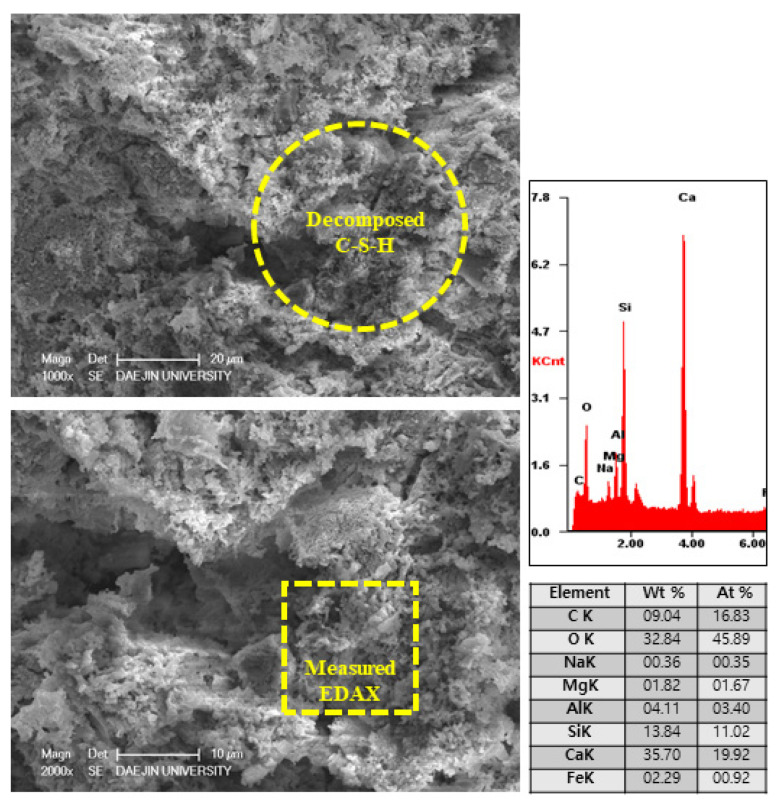
SEM-EDAX results of P+SiC sample after internal temperature experiment (1000 °C).

**Figure 23 materials-15-01959-f023:**
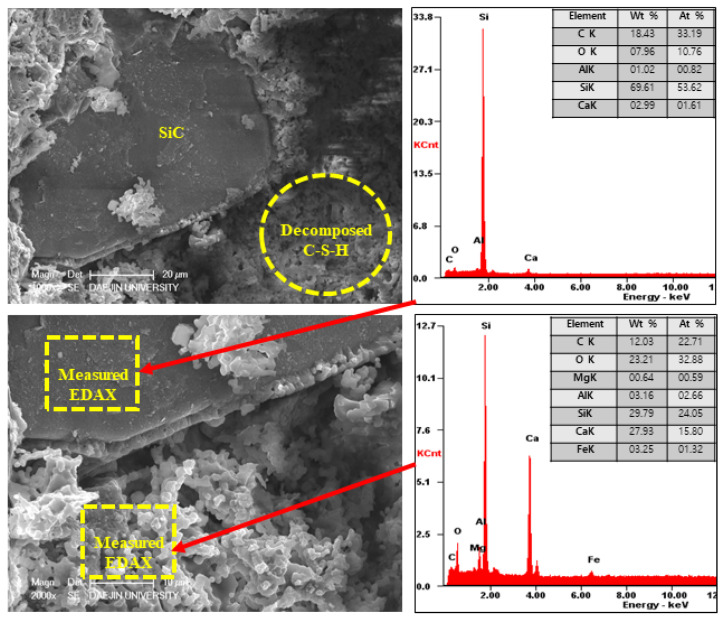
SEM-EDAX results of EA100 sample after internal temperature experiment (1000 °C).

**Figure 24 materials-15-01959-f024:**
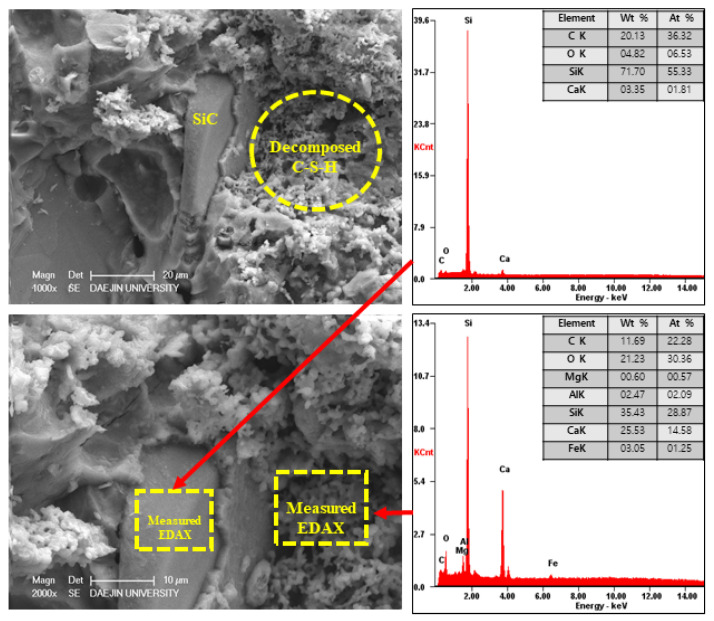
SEM-EDAX results of EA100+SiC sample after internal temperature experiment (1000 °C).

**Table 1 materials-15-01959-t001:** Concrete mix proportions. Reprinted from [13].

Type	W/B (%)	S/a (%)	Water (kg)	Cement (kg)	Fine Aggregate (kg)	SiC (kg)	Coarse Aggregate (kg)	EA (kg)	Water Reducer (kg)
P	44.9	45.8	180	401	766	-	914	-	2.01
EA30							640	274	
EA50							457	457	
EA70							247	640	
EA100							-	914	
P+SiC					727.7	38.3	914	-	
EA30+SiC							640	274	
EA50+SiC							457	457	
EA70+SiC							274	640	
EA100+SiC							-	914	

## Data Availability

Not applicable.

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
