# Peer review of "Investigation of Newly Developed PCM/SiC Composite Aggregate to Improve Residual Performance after Exposure to High Temperature"

_materials, 2022, doi:10.3390/ma15051959_

Round 1

Reviewer 1 Report

  1. Page 2 line 52, There is a spelling mistake
  2. The definition and compositions of PCM/SiC composite aggregates should be given in Abstract and Introduction part respectively so that readers can understand rapidly and clearly. The research significance should be given in the Introduction.
  3. Page 2 line 70, the reason that PCM/SiC composite aggregates equals to EA should be illustrated. Is PCM without SiC called EA as well in Talble 4?Please revise.
  4. Page 3 line 111, for PCM/SiC composite aggregates as the coarse aggregate, the fabrication seems to be complex, how about the economy of concrete with it? The application scenario should be clearly given.
  5. Page 4 line 112, the basis of mix proportions should be given, such as the selection of water-binder ratio, sand-coarse aggregate ratio, and the calculation of water reducer.
  6. Page 5 line 159, The working mechanism of PCM improving the high thermal resistance of concrete should be supplemented.
  7. Page 6 line 176-200, the compressive strength of specimens EAs was decreased in comparison with plain specimens. Does this affect its application? Please explain.
  8. Page 6 line 192, For concrete with EA, how ITZ affects the reduction of mechanical properties? ITZ should be analyzed to demonstrate it.
  9. Part 3.1.3, DTG/TGA Analysis should be more deeply, it is encouraged to conduct some quantitative analysis.
  10. Page 10 line 269, the reason that ettringite peak was disappeared after 7 days while re-appeared after 28 days should be analyzed, and it seems unreasonable. On the other hand, the reason that C-S-H content in concrete with EA100 was higher than that in plain concrete should be given as well.
  11. Page 14 line 327, SiC particles (green) around aggregate should be marked in Figure 18.
  12. Page 15 line 363, the phase of paraffin wax changes at temperature around 40℃ and the flash point is 200℃, that means phase of paraffin wax changes from solid to liquid. The more EAs, the more liquid in concrete after 200℃ is, and the lower compressive strength is. Therefore, what is the relationship between the higher residual compressive strength and improved thermal resistance properties?

Author Response

The authors are thankful to the reviewers for their valuable comments and suggestions. The reviewers’ comments helped improve the quality of the manuscript. 

Reviewer 2 Report

Most of the figures showing the results from mechanical testing should be labeled with the curing time. It is assume that the test were done after 28-day curing. It is not totally clear when the specimens were heated. Please explain.

It seems that you really have two experiments, one in which you replace the aggregate with EA and another in which you heat the material to high temperatures. It would be helpful if your conclusions separated those two experimental conditions.

Author Response

(The authors gave the same response as above.)

Reviewer 3 Report

Very interesting article; recommended for publication but with significant changes;

Must be improved:

  • Language: examples:

                   Line 124 “ Cylindrical concretes with sized 100 x 200 mm3.. “

                    Line 128 “ Scanning Electron Microscope was analyzed by SEM equipment..”

etc.

  • description of figures, (examples: fig.3 “result”? fig. 5,”Sic” fig. 7?
  • descriptions of tests results
  • quality of the drawings – i.e. Figure 7,

- Does fig. 2 present the same samples before and after heating?

- Fig. 6 – is there any correlation between the compared results?

- 3.1.5 SEM-EDAX Analysis and p. 4.2.5 SEM-EDAX Analysis?

Can the authors coment on taking such areas for E-DAX analysis, especially as shown in Figures 12 and 13 (SEM Images with E-DAX analysis) – in my opinion the E-DAX results don't give  the information about the tested samples properties)

Author Response

(The authors gave the same response as above.)

Round 2

Reviewer 1 Report

-

Author Response

Thank you for your review.

Reviewer 2 Report

Authors have addressed my concerns

Author Response

We are very grateful that you reviewed our paper. Thank you very much.

Reviewer 3 Report

Extensive editing of English language and style required.

Author Response

(The authors gave the same response as above.)
